Urinary microRNA can be concentrated, dried on membranes and stored at room temperature in vacuum bags

Zhang Fanshuang 1
Cheng Xiaoyu 1
Yuan Yuan 1
Wu Jianqiang 1
Gao Youhe 1 2 gaoyouhe@bnu.edu.cn
1 Department of Pathophysiology, Institute of Basic Medical Sciences Chinese Academy of Medical Sciences, School of Basic Medicine Peking Union Medical College , Beijing , China
2 Department of Biochemistry and Molecular Biology, Gene Engineering and Biotechnology Beijing Key Laboratory, Beijing Normal University , Beijing , China
Soares Paula
Electronic publication date: 2015 Jul 14
Publication date: 2015
Volume: 3
Electronic Location ID: e1082
Received 2015 Feb 28; Accepted 2015 Jun 15
Copyright: © 2015 Zhang et al.
Copyright year: 2015
Copyright holder: Zhang et al.
License: This is an open access article distributed under the terms of the Creative Commons Attribution License, which permits unrestricted use, distribution, reproduction and adaptation in any medium and for any purpose provided that it is properly attributed. For attribution, the original author(s), title, publication source (PeerJ) and either DOI or URL of the article must be cited.
License URL: https://creativecommons.org/licenses/by/4.0/

Keywords: Urinary microRNA, Concentration, Preservation, Nylon membrane, Room temperature

Funding: National Basic Research Program of China 2012CB517606 2013CB530805 Key Basic Research Program of the Ministry of Science and Technology of China 2013FY114100 111 Project B08007 PUMC Youth Fund Fundamental Research Funds for Central Universities 33320140133 This work was supported by the National Basic Research Program of China (2012CB517606, 2013CB530805), Key Basic Research Program of the Ministry of Science and Technology of China (No. 2013FY114100), 111 Project (B08007), PUMC Youth Fund and the Fundamental Research Funds for the Central Universities (33320140133). The funders had no role in study design, data collection and analysis, decision to publish, or preparation of the manuscript.

==============================
Urine accumulates traces of changes that occur in the body and can potentially serve as a better biomarker source. Urinary microRNA is a promising class of non-invasive disease biomarkers. However, long-term frozen human urine samples are not a good source for the extraction of urinary microRNA. In this paper, we demonstrate that urinary microRNA can be concentrated, dried on membranes and stored in vacuum bags at room temperature for several months. The amount of total RNA on the membranes after storage at room temperature for three months was unchanged. The levels of miR-16 and miR-21 exhibited no significant differences (P = 0.564, 0.386). This simple and economical method makes the large-scale storage of clinical samples of urinary microRNA or other nucleic acids possible.

Introduction

The fundamental property of a biomarker is change. Whereas plasma typically maintains a homeostatic internal environment, urine tends to reflect changes occurring inside the body. This property potentially makes urine a better biomarker source than plasma (Gao, 2013; Li, Zhao & Gao, 2014). In addition, urine can be obtained in large quantities using non-invasive procedures, and various components of urine are relatively stable given that they have been incubated in the bladder at 37  °C for several hours.

Changes in nucleic acids are a promising biomarker for diagnoses, prediction and monitoring of diseases (Ralla et al., 2014; Veltri & Makarov, 2006). MicroRNA, which belongs to the class of small non-coding RNAs, functions by base-pairing with mRNA molecules, thereby resulting in the silencing of these mRNAs (Bartel, 2004). Currently, 1,193 precursors and 1,915 mature microRNA sequences have been reported in mice (miRBase 21.0). Similarly, 1,881 precursors and 2,588 mature microRNA sequences have been reported for humans (miRBase 21.0). MicroRNA has no known post-processing modifications; thus, the composition is less complex than other biomolecules (Weber et al., 2010). For this reason, microRNA is a promising new disease biomarker for ailments such as malignancies of the prostate, bladder and kidney as well as other urologic diseases which have already been extensively studied and reviewed (Catto et al., 2011; Mlcochova et al., 2014; Ralla et al., 2014; Yang et al., 2013; Zhang et al., 2014). The preservation of urinary microRNA is the foundation of the application of urinary microRNA as the biomarker source for diagnoses. In 2013, the stability of microRNAs in urine and urine storage conditions were evaluated by Mall et al. (2013) and this study demonstrated that microRNA was relatively stable in the harsh urinary environment, even at varied temperatures and after ten freeze–thaw cycles. However, the median total RNA concentration of urine is 94 µg/L (with 129 interquartile range) in contrast to concentrations of 308 µg/L (with 104 interquartile range) in plasma and 47,240 µg/L (with 73,180 interquartile range) in breast milk (Weber et al., 2010).

Recently, a new type of commercial urine preservation tube was invented for the preservation of urine samples. However, the tubes are quite expensive, and this simple preservation method without concentration requires a significant amount of space. Jia et al. (2014) introduced a method for directly adsorbing urinary proteins onto polyvinylidene difluoride (PVDF) membranes that can be subsequently dried and stored; however, other urinary components were not considered. Based on this method, an alternative nylon membrane, which exhibits stronger binding affinities for nucleic acids including DNA and RNA via electrostatic interaction, was assessed for its ability to concentrate and preserve urinary microRNA.

Materials and Methods

Ethics statement

The purpose of this paper is to examine the use of nylon membranes to preserve and concentrate urinary nucleic acids. Collecting urine does not harm the donor at all. Instead of using individual urine samples, mixed urine samples were used in this study, so personal information did not documented. The specimens were collected after obtaining verbal informed consent from the participants or written consent from the participants who were willing to sign. The consent procedure including verbal or written consent and research protocol were approved by the Institute of Basic Medical Sciences Medical Ethics Committee of Peking Union Medical College (Project No. 040-2014). All of the records of consent by the participants were documented by the authors.

Urine collection

Urine samples from 10 healthy participants were collected and stored at 4 °C and subsequently combined. All of the urine samples were centrifuged at 5,000 × g for 30 min at 4 °C. The supernatant was aliquoted into 50 mL sample tubes and stored at −80 °C until use. The participants included four males and six females ranging from 24 to 30 years in age.

Urinary nucleic acids preservation on the membranes

The experimental procedure was performed as described by Jia et al. (2014) with modifications. In Step 1, the 0.45-µm Positively-Charged Nylon 6,6 membranes (Cat# BNBZF810S, Pall, NY) and the medium-speed qualitative filter papers were cut according to the diameter of the vacuum suction filter bottle (10 cm2 filter area). In Step 2, three to four sheets of wet filter paper were placed onto the vacuum suction filter bottle, and one wet nylon membrane was placed immediately onto the filter papers to avoid the generation of bubbles. In Step 3, the vacuum suction filter bottle was installed and connected to the vacuum pump. In Step 4, the collecting bottle was then loaded with urine supernatant. In Step 5, the vacuum pump was opened to allow the urine to pass through the nylon membrane in a drop-wise fashion. In Step 6, the intensity of the vacuum pressure was set to approximately 5 kPa with an initial velocity of approximately 1 to 2 drops per second. As time passed, the speed of the drops was reduced, so a higher intensity vacuum pressure was required. The total filtration process time was 5 to 10 min, and the intensity of the vacuum pressure was 5 to 50 kPa, depending on the volume of the urine. In Step 7, after the urinary nucleic acids were adsorbed onto the nylon membranes, the membranes were placed into the 56 °C drying oven for 3 to 5 min to complete the drying process or left to dry at room temperature. In Step 8, the dry membranes were placed between aseptic sealing bags and then sealed using a vacuum sealer. Tags were added to the dry membranes that contained the basic information on the participants and the unique number of the membranes, which could be cross-referenced with additional information relevant to the urine sample. Finally, the membranes were stored at −80 °C or room temperature. A flow chart explaining the process is presented in Fig. 1.

Figure 1 The scheme of microRNA concentration and preservation.

Urinary RNA extraction

RNA is more labile than DNA and ribonuclease (RNase) is known to be present in the body fluids (Reddi & Holland, 1976; Zhao et al., 2015). Consequently, RNA molecules on the urinary nucleic acid-bound dry membranes were examined in this paper.

The urinary nucleic acid-bound dry membranes were cut into small pieces and placed in clean tubes. Total RNA, including microRNAs, was extracted by using TRIzol reagent (Invitrogen, Carlsbad, CA) according to the manufacturer’s instructions. Briefly, 250 µL of chloroform was added, and the samples were shaken vigorously for 15 s. Then the samples were centrifuged at 12,000 rpm for 15 min at 4 °C. The fraction from the top aqueous phase was obtained and transferred into new 1.5-mL tubes. Next, 600 µL of isopropanol was added, and the solution was centrifuged for 15 min at 12,000 rpm at 4 °C. After removing the aqueous solution, 1 mL of 75% ethanol was added, and the sample was centrifuged for 10 min at 12,000 rpm at 4 °C. After removing the ethanol, the pellet was dissolved in RNase-free water and quantified with a NanoDrop spectrophotometer (Thermo Fisher Scientific). Finally, the sample was stored at −80 °C for further analysis by RT-qPCR.

MicroRNA quality control

The quality of the microRNA was evaluated on a Bioanalyzer 2100 instrument (Agilent, Santa Clara, CA, USA) using small RNA kit.

Reverse transcription of urinary microRNA

U6, miR-16 and miR-21 were reverse transcribed using synthetic primers and the Promega GoScript™ Reverse Transcription kit (Promega Corporation, Madison, USA) according to the manufacturer’s protocol. After combining the experimental primers with an equal amount of template RNA, the primers and template mix were thermally denatured at 70 °C for 5 min and chilled on ice for 5 min. The primers and template mix were added to the reaction mix on ice. Following an initial annealing at 25 °C for 5 min, the reaction was incubated at 42 °C for up to one hour. Then, the reaction was incubated in a controlled-temperature heat block at 70 °C for 15 min. The reaction product was placed on ice or stored at −20 °C until use.

Real-time PCR of urinary microRNA

To quantify urinary microRNA, real-time PCR amplification was performed in a 20-µl reaction volume on a Bio-Rad real-time PCR system according to the manufacturer’s protocols. The real-time PCR reactions were set up in triplicate in 8-tube PCR strips (AXYGEN, Union City, CA 94587, USA) and briefly centrifuged. Real-time PCR conditions included one cycle of initial activation for 10 min at 95 °C followed by 40 cycles of denaturation for 15 s at 95 °C, annealing for 10 s at 95 °C and extension for 30 s at 55 °C. The melting program was performed at 55 °C to 94.5 °C at a heating rate of 1 °C per 5 s. The levels of the target microRNAs were normalized using U6, and the relative level of microRNA was determined using the ΔΔCt method. All of the samples were tested in triplicate.

Statistical analysis

All statistical analysis was performed with the SPSS software (version 16, IBM), and all tests were two-sided with a 0.05 significance level. The total RNA amounts on the membranes before and after three months of storage at room temperature were compared using t-tests. The CT values of microRNA before and after three months of storage at room temperature were compared using Mann–Whitney U tests to account for differences in variance between groups.

Results

Loading capacity of the nylon membranes

Centrifuged urine (100 mL) was filtered through a 10-cm2 nylon membrane, which adsorbs urinary nucleic acids by electrostatic interaction, and the filtrates were then filtered through a new nylon membrane to detect urinary nucleic acids not adsorbed by the first membrane. The urinary nucleic acid-bound membranes were dried completely in a 56 °C drying oven, and RNA was extracted and quantified. Experiments repeated in triplicate indicated that the average amount of RNA extracted from the first nylon membrane was 4,442.1667 ± 333 ng, whereas 759.1 ± 21.83 ng of RNA was extracted from the second nylon membrane. A comparison of the nucleic acid adsorption amounts from the first and second membranes revealed that most of the RNA was adsorbed on the first membrane. The binding rate of urinary RNA was 85.41% [4,442/(4,442 + 759)].

Reproducibility of urinary RNA preservation

The adsorption of 50 mL of mixed centrifuged urine on 10 cm2 nylon membrane was repeated 7 times. Then, the membranes were placed into the 56 °C drying oven for 5 min to complete the drying process. The total RNA that was adsorbed onto each nylon membrane was extracted and quantified with a NanoDrop spectrophotometer. As shown in Table 1, the average amount of total RNA from 7 repeats was 7,788.65 ± 145.9 ng, and the coefficient of variation was 0.0187.

Table 1 Total amount of RNA.

No.	RNA amount, ng	
1	7,669.35	
2	7,894.40	
3	7,832.15	
4	7,832.00	
5	7,825.67	
6	7,947.00	
7	7,520.00	

The amount of urinary microRNA stored at room temperature for three months was unchanged

Adsorption of 100 mL of mixed centrifuged urine on 10 cm2 nylon membranes was repeated 8 times. Then, the membranes were placed into the 56 °C drying oven for 5 min to complete the drying process. Four sheets of membranes were chosen randomly, and total RNA adsorbed onto the nylon membranes was extracted and quantified using a NanoDrop spectrophotometer (Table 2). The other four membranes were stored at room temperature for approximately 3 months. Total RNA adsorbed onto the nylon membranes was then extracted and quantified with a NanoDrop spectrophotometer (Table 2). The average amount of the total RNA from 4 independent experiments from membranes before room temperature storage was 11,264.70 ± 666.23 ng, and the coefficient of variation was 0.0232. The average amount of total RNA after 3 months of room temperature storage was 11,920.73 ± 766.56 ng, and the coefficient of variation was 0.0219. No significant difference was observed before and after storage (P = 0.244). MicroRNA quality stored at room temperature for three months was unchanged (Fig. 2). U6, miR-16 and miR-21 were also quantified by qRT-PCR as described in the ‘Materials & Methods’ section. Relative CT values were proportional to urinary microRNA quantity (Mall et al., 2013), and U6 was reported as an internal normalization control (Mlcochova et al., 2014). Our data indicate that U6 was detectable in the urine of all of the 8 samples, and no significant difference in CT values (P = 1.000) was observed before and after storage. Additionally, the miR-16 and miR-21 CT values did not significantly differ before and after storage (Fig. 3).

Table 2 Total amount of RNA before and after storage.

No.	RNA amount, ng (before storage)	No.	RNA amount, ng (after storage)	
1	10,853.10	5	11,370.33	
2	11,867.70	6	11,173.53	
3	11,793.60	7	12,396.78	
4	10,544.40	8	12,742.29	

Figure 2 (A) Gel electrophoresis analysis of small RNA fractions on an Agilent small RNA chip. (B) Fluorescence intensity of small RNA fractions at various sizes (nt).

Lane 1–4 represented the membranes before 3 month storage. Lane 5–8 represented the membranes after 3 month storage.

Figure 3 Scatter plots of urinary levels of miR-16 (A) and miR-21 (B) in the before storage group (n = 4) and the after storage group (n = 4).

MicroRNA levels (Log10 scale at Y-axis) were normalized to U6. The line represents the median value.

Discussion

MicroRNA has been proposed as a promising class of biomarkers (Kroh et al., 2010; Mall et al., 2013; Mitchell et al., 2008). This study exclusively examined total RNA and microRNA levels on the urinary nucleic acid-bound membranes.

Although it has not been widely used until recently, urinary DNA is a potential source of material for molecular diagnosis (Chan et al., 2003; Goessl et al., 2000; Zhang et al., 1999). However, the storage of fresh urine at temperatures of 4 °C or lower results in significant DNA degradation (Hilhorst et al., 2013; Yokota et al., 1998). To overcome this disadvantage, a magnetic bead-based method was developed for extracting and concentrating DNA from urine (Bordelon et al., 2013). However, this method only utilizes samples of urine up to 5 mL, and an increase in volume to as high as 20 mL may require a longer adsorption step (Bordelon et al., 2013). According to the property described by the company or other articles (Kainz, Seifriedsberger & Strack, 1989; Khandjian, 1986; Taylor, 1985; Thurston & Saffer, 1989), “It offers high sensitivity in nucleic acid detection applications. The binding mechanism for membrane is through electrostatic interactions” (Pall Corporation, 2015; http://www.pall.com/main/oem-materials-and-devices/product.page?lid=gri78lty). All kinds of nucleic acids including DNA and RNA can all be bound on membranes. Based on this, we proposed that the method can also be used for DNA concentration and preservation too. What’s more, these urinary nucleic acid-bound dry membranes may be stored considerably longer than three months.

Generally, 200 μL or 400 μL fresh urine will be used for each RNA extraction. Because of the low concentration of urinary RNA, it often costs several extractions to have enough RNA for reverse transcription. The method described here provides an economical way to concentrate and preserve urinary RNA. Different urine samples may have different amount of RNA on the membrane. However, no matter how much RNA could be recovered, as long as the amount of templates used for PCR reaction remained the same, the target microRNAs can still be quantified and compared.

Urine is a better source for biomarker research (Gao, 2013). The current study was the first to report the use of nylon membranes to concentrate and preserve urinary nucleic acids. The nucleic acids preserved on nylon membranes could be extracted and subsequently analyzed using traditional downstream analytical applications, such as PCR, real time PCR, nucleic acid microarray or other nucleic acid test methods. This method allows for the preservation of urine samples from all consenting patients during every stage of disease development or even for those consenting healthy individuals for prospective studies.

Supplemental Information

Table S1 Raw data

The CT values of U6, miR-16 and miR-21 before and after storage.

Click here for additional data file.

Additional Information and Declarations

Competing Interests

Author Contributions

Human Ethics

The authors declare there are no competing interests.

Fanshuang Zhang conceived and designed the experiments, performed the experiments, analyzed the data, contributed reagents/materials/analysis tools, wrote the paper, prepared figures and/or tables.

Xiaoyu Cheng researched the overestimation of the concentration of urinary RNA extracted using miRNeasy kit and TRIzol reagent.

Yuan Yuan contributed reagents/materials/analysis tools.

Jianqiang Wu analyzed the data.

Youhe Gao conceived and designed the experiments, reviewed drafts of the paper.

The following information was supplied relating to ethical approvals (i.e., approving body and any reference numbers):

Institute of Basic Medical Sciences Medical Ethics Committee of Peking Union Medical College—Project No. 040-2014.

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
