# Peer review of "Urinary microRNA can be concentrated, dried on membranes and stored at room temperature in vacuum bags"

_PeerJ, doi:10.7717/peerj.1082_

## Round 0.1 · original submission · Major Revisions

Please consider and reply to all the concerns raised by the reviewers. The title and some sentences in the text extrapolate the experimental data presented, please reformulate being strict to the results obtained. No comparison has been made with nucleic acids isolated from fresh urine and, consequently, what fraction of urinary nucleic acids are captured on the membrane is unknown, please clarify this aspect. Please explain more precisely the interest of the method in the context of human diseases.

Reviewer 1 ·

Basic reporting

The present finding is the RNA or microRNA isolation from the urinary sources, requiring further clarification. Urine contains valuable molecules as the biomarker sources for the diseases diagnosis. Nucleic acids are also classified for the biomarkers, as the authors isolated the urinary nucleic acids using membranes with long term storage. They claims that the present method is simple convenient and economical for the large-scale storage of urinary nucleic acids.
Without any application to functional meaning or scientific solution in human diseases or fundamental biology, it is a marginal interest. Thus, I suggest the further scientific extension should be progressed using their isolated urinary RNAs with regard to the biological functions such as diseases makers. Or I can recommend that the MS can be submitted to more specialized journals such as Biotechniques and Methods, etc.

Experimental design

The experiment the authors conducted is good. But more extensive experiment results are requred for the justification.

Validity of the findings

The information including the sequential structures, specific formation of RNAs in patients and normal individuals, relationship between diseases status and diversity in Nucleic acids is interesting and valuable for further clarification. However, the present form is to marginal.

Additional comments

Further extensive studies are needed.

Reviewer 2 ·

Basic reporting

The paper reports a method to recover cell-free micro-RNAs from urine samples filtered through nylon membranes. The paper makes a well substantiated case thar urinary RNA and micro-RNAs can be collected and stored by the method herein described and the paper is clearly written in this sense. However I find that the title's expression "Urinary nucleic acids..." is not adequate since DNA or messenger RNA were not assessed in thus work. Thus I would suggest that title was more strict regarding the actual experiments performed and could be corrected to: "Urinary micro-RNAs...".The paper includes sufficient background of prior literature. I would suggest to merge tables 2 and 3.

Experimental design

The manuscript describes adequate experiments to validate the method under evaluation. Further experimental work regarding DNA and messenger RNA could have been performed in order to demonstrate a broader scope of the method.Methods are described in sufficient detail.

Validity of the findings

Data seems credible and robust as a number of technical replicates were performed. The paper concludes that this method's feasibility could be extrapolated for all types of nucleic acids (sentence in line 160-161). However experimental demonstration for this statement is lacking and the level of integrity and feasibility to amplify other nucleic acids such as DNA and messenger RNA would be needed to validate this conclusion.

Additional comments

The paper makes a solid case regarding the capture and feasibility for the analysis of micro-RNAs. The title and the conclusion in lines 160-161 overstate and broaden the works scope without supporting experimental data. If authors limit the scope of the work to micro-RNAs and change title accordingly it is my opinion that the paper is suited for publication.

---

## Round 0.2 · Minor Revisions

Please address in the revised text the remaining comments of Reviewer 1. Please change the legend of Figure 1 "The flow chart of the method." since it shows a scheme of the filter apparatus instead of the scheme (flow chart) of the method.

Reviewer 1 ·

Basic reporting

The authors have revised the manuscript.
Although the study scientifically sounds, the careful approaches including microRNA sequences and application are required to justify the present method description from the urine sources.

Experimental design

Some more specific analysis datasuch as the isolated microRNA sequences and their application to human diseases are appreciated.
Previously, the exsitence of microRNAs are know from the urine.

Validity of the findings

The application of the micro RNAs are important. The present data show the identification and electrophoretic resolution only.

Additional comments

More extensive and scientific analysis using the isolated microRNAs are apreciated after revision.

Reviewer 2 ·

Basic reporting

The paper reports a method to recover cell-free micro-RNAs from urine samples filtered through nylon membranes. The paper is objective and makes a well substantiated case that urinary RNA and micro-RNAs can be collected and stored by the method herein described. The paper is clearly written in this sense. The paper includes sufficient background of prior literature. Figures and tables are self-explanatory.

Experimental design

The manuscript describes adequate experiments to validate the method under evaluation. Further experimental work regarding DNA and messenger RNA could have been performed in order to demonstrate a broader scope of the method.Methods are described in sufficient detail.

Validity of the findings

Data seems credible and robust as a number of technical replicates were performed. The paper concludes and validates that this method is feasible for the analysis of urinary micro-RNAs.

Additional comments

The paper makes a solid case regarding the capture and feasibility for the analysis of micro-RNAs. The title, the supporting experimental data and the conclusions of the paper are adequate within this scope. Given that authors have limited the scope of the work to micro-RNAs it is my opinion that the paper is suited for publication.

---

## Round 0.3 · accepted · Accept

No further comments to the manuscript